# Harnessing Systems Science and Co-Creation Techniques to Develop a Theory of Change towards Sustainable Transport

**Caitriona Corr ***, **Niamh Murphy and Barry Lambe**

Department of Sport and Exercise Science, South East Technological University, X91 HE36 Waterford, Ireland;
niamh.murphy@setu.ie (N.M.); barry.lambe@setu.ie (B.L.)
* Correspondence: caitriona.corr@postgrad.wit.ie

**Abstract:** Integrated transport and land-use planning and reduced car dependency proffers a pathway to mobility justice and reduced transport poverty, whilst providing opportunities for potential health benefits and reducing carbon emissions. In spite of this, smaller cities and towns face opposition to the reallocation of road space away from the private car. Traditionally, transport measures have been responsive to growing car-use. To accelerate a behavioural shift to sustainable modes, an evidence-based, transformative approach is required that is consultative in nature, and inclusive of all relevant stakeholders and nurtures innovation. The study aims to achieve this by co-creating a theory of change, with a broad group of stakeholders and the community. Systems science and co-creation techniques were utilised to enable informed decision-making and foster shared learning, resulting in a theory of change formulated by stakeholders with a shared vision. Sixteen objectives were identified under five broad categories, create sustainable systems; design healthy built environments; engage society; empower people and prioritise road safety, informed by the systems-based framework Global Action Plan on Physical Activity. Assumptions, risks and key performance indicators were key elements of the theory of change. Risks identified for successful implementation of the plan were lack of funding and human resources to deliver actions, political challenges, lengthy planning processes, entrenched social norms and resistance from the community. This process, which was adopted, synthesises scientific evidence, a participatory systems approach, informed decision making and the practical application of the embedded researcher, resulting in a pragmatic theory of change to reduce car-dependency and create a shift to sustainable travel modes. The process highlights the importance of stakeholder and community engagement, from participatory mapping of the system to the development of the theory of change to generate local solutions to identified challenges. The resulting theory of change will form the basis of a Sustainable Urban Mobility Plan for Kilkenny City. The theory of change can be adapted to new settings by the participatory processes outlined.

**Keywords:** sustainable urban mobility; active travel; systems; theory of change; stock and flow analysis; sustainable cities

## 1. Introduction

The necessity to create a modal shift to sustainable transport in response to the urgent need to reduce carbon emissions is well documented [1–3]. Cycling, together with walking, scooting and wheeling (use of mobility aids), has the additional environmental benefits of improved air quality and reduced noise pollution [4,5]. The reduced spatial requirements of active travel and collective transport modes enable more appropriate and fairer use of public space and allow for additional greening opportunities, sustainable urban drainage systems, provision of seating and mobility access, retail uses and access to public transport or share schemes [6]. The provision of cycling infrastructure reduces the financial burden of transport to society [7]. Furthermore, a modal shift to cycling contributes to reduced car-dependency, thereby ameliorating transport poverty and bettering accessibility to essential services for all. The health benefits of increased cycling levels in the community include

reduced road traffic injuries and deaths, reduced incidence of non-communicable diseases, mortality, and enhanced well-being [8,9]. Cycling, and other active modes, encourage social interactions and enrich social capital [10,11]. Cycling is a more equitable transport mode that enables independent journeys for those who are completely dependent on car-centric urban systems, i.e., children and young people, people with reduced cognitive function or physical impairments that preclude them from driving, older adults, or those that cannot afford a car. However, to create the physical and societal space for a shift towards cycling, a step change is required in the urban fabric. Solutions need to go beyond focusing on cycling to consider cycling as part of the entire mobility system. Mobility as a system overlaps and is intertwined with other systems such as environment, health, tourism, politics and the economy and therefore, a transformative systems approach is required.

Across Europe, the response to COVID-19, coupled with the energy crises exacerbated by the war in Ukraine, accelerated the reallocation of space towards cycling infrastructure [12]. Initially, the response was more evident in larger cities, with many advancing existing plans. The need to create car-free urban centres and shift to sustainable modes in smaller urban areas has been more challenging. In Ireland, cycling for transport across towns and smaller cities has declined drastically since the 1990s from 13% to 3% [13]. As is typical of low cycling countries, of those cycling, sectors of the population are under-represented including females, children, secondary school students and older adults. Safety is cited as the main barrier for those who would prefer to cycle, and segregated, connected cycling infrastructure is lacking in smaller urban centres. Motivators that exist in larger urban areas, such as congestion and convenience, are absent in lower-density and dispersed populations. Private cars are viewed as the most convenient, comfortable and fastest mode of transport and car ownership remains a strong aspiration. The lack of, or low-attractiveness of other mobility solutions results in forced car ownership to maintain access to social, health, education and employment opportunities and avoid transport disadvantages [14]. Over the last number of decades, the greater allocation of space to private cars has led to the creation of hostile environments for pedestrians and cyclists. This induces greater car use and car dependency and by favouring the car, urban sprawl is enabled, thereby further reducing the potential for active modes [15]. Decades of underinvestment in public transport and active travel has left these modes as unattractive options accounting for small mode shares. Development of new infrastructure is a lengthy process and does not impose restrictions on car travel. The reallocation of car-orientated spaces will combine push and pull factors to accelerate a shift to sustainable transport. We must move away from car-centric design and reallocate space for people focusing on the approaches identified under the Climate Action Plan to avoid (transport demand) and shift (to sustainable modes). This requires new skill sets such as a shift in decision making, for politicians and town planners, transformative road design for engineers and design teams and a sea change in public behaviour. Innovative collaboration, planning and delivery mechanisms are required to bring about the necessary change.

Previous research conducted in Kilkenny City, in the southeast of Ireland, examined the factors that led to a reticence to use of car-restrictive policies and found the power of the trader lobby, together with the failure to engage the silent majority in the wider community resulted in the underuse of car-restrictive policies. Key recommendations were the use of a comprehensive suite of measures and community engagement as utilised in the Living Streets and Ciclovias projects [16]. Similarly, an Inter-Reg European Peer review in 2021 recommended a broad community and stakeholder engagement to generate support for transformative actions [17]. Therefore, the formative research for this study adopted a systems approach to consider the inter-relationships of the mobility system, and the stakeholders involved. Through community-wide surveys, semi-structured interviews, workshops with community organisations and focus groups with under-represented cohorts of the population, the attitudes, barriers to, and enablers of cycling and mobility in the city were explored [18]. The findings of this study shifted the framing of this research. The lack of community and stakeholder engagement and the resulting lack of trust and under-

standing of roles was seen as a key barrier to the success of plans. A collaborative process with broad stakeholder engagement to encourage vision-orientated and community-based decision-making was fundamental to succeed in transforming the wider mobility system. The Physical Activity through Sustainable Transport Approaches (PASTA) consortium [19] reviewed 26 published frameworks and combined behavioural concepts, structural features and a large number of determinants in a single framework. However, the study concluded that large research projects may still merit a study-specific framework. This allows for adaptation to local contexts and the local socio-ecological system, but crucially, through stakeholder involvement, it allows for innovation and learning. In response to this, in 2022, this research adopted a participatory systems approach, to co-create a theory of change towards accelerating a shift to sustainable modes.

A Theory of Change, (ToC) may be defined as "a rigorous yet participatory process whereby groups and project stakeholders identify the conditions they believe have to unfold for their long-term goals to be met" [20]. The conditions in the ToC are modelled as inputs, activities or interventions, outputs, outcomes, and impacts that are arranged in a causal framework. The participatory process helps align different stakeholders to the intended impacts and helps them understand their roles. It can also highlight ways that interventions might need to be executed differently, or sequenced with other interventions, to maximise the chances of success and minimise risk [21]. Explicit description of the logic for change also makes ToC approaches particularly suited to process evaluation, investigating to what extent the activities and intermediate steps of an intervention are happening, before real impacts would be observable [22]. This allows early recognition of whether interventions are working or whether adaptations may be needed, thereby reducing the chances of failure. Furthermore, the process strengthens the effectiveness of institutions and mechanisms based on a collaborative and participatory process, thereby instigating sustainable and inclusive changes [23]. Originally used most in the community initiative and international development sectors [24], it is now widely used as a tool for project/programme planning and evaluation and policymaking. Increasingly, ToC is used to address complex, societal challenges [25,26], in some instances resulting in immediate adoption of final research outcomes [27]. The MOTION project has identified the ToC as a key framework to support the widespread adoption of new transformative, systems innovation policies and practices, specifically in land use, urban planning and mobility [28]. In Kilkenny City, the ToC forms the basis of a Sustainable Urban Mobility Plan for the city, led by Kilkenny County Council but with the intended outcomes embedded in the narrative of all organisations involved.

As a preparatory step, to facilitate informed decision-making, two system science techniques were chosen to conduct a rigorous analysis of the previously mapped mobility system; causal loop construction and stock and flow analysis. The results of these two steps were presented to the stakeholders and community throughout the co-creation process. The former technique explores interrelationships and helps identify existing vicious circles and leverage points. The latter investigates the accumulation of stocks in a system and the flows that affect them over time such as existing resources, services and investment streams. This analysis results in a synthesis of the financial information required to sense-check the deliverability of interventions. It also provides an accurate portrayal of the mobility services available, the cost of provision of services (where available) and the take up of services. Both techniques assist in progressing interventions from idea generation to implementation.

This paper describes the steps involved in advancing from a participatory systems approach to a co-created, pragmatic theory of change. Firstly, the review of the systems map conducted by the researcher using systems science techniques is described. Secondly, the process of establishing the platform for the co-creative process is discussed. Thirdly, the intervention design is described and finally, the resultant theory of change is presented. This process has resulted in the development of a vision, objectives and interventions for a Sustainable Urban Mobility Plan (SUMP) for Kilkenny City. In Kilkenny City, the SUMP process will continue to refine and shape the interventions to fit the local context

and resources, through ongoing stakeholder engagement. The co-created theory of change presents a suite of interventions, easily adapted to towns and small cities, to accelerate a shift to sustainable transport modes.

*Study Area and Context*

Kilkenny City is a medieval city in the southeast of Ireland, with a population of 27,184 and a population density of 2348 persons per square kilometre [29]. It is the main urban centre, in the predominately rural county of Kilkenny (population, 104,160). It is the eighth largest employment centre in the state and is a self-sustaining urban centre. It is a predominately flat and compact city, that straddles the River Nore, with a network of medieval lanes and narrow streets in the city core.

## 2. Methodology

### *2.1. Research Design*

This research study consisted of a series of systematic steps designed to develop a co-creation process to create a ToC. These steps, related activities and how they are aligned are illustrated in Figure 1 below. The formative research resulted in a systems map of the factors influencing cycling in Irish towns and the wider mobility system. This process incorporated the findings of the literature review to identify variables influencing cycling in smaller cities and towns, the learnings of the researcher working alongside the city's active travel team for an 18-month period and the findings of a stakeholder engagement, comprising a community-based survey (n = 437), semi-structured interviews (n = 7), workshops (n = 3) and focus groups (n = 7). The systems map is presented in Supplementary Materials S1, while the complete methods and results are detailed in a previous publication [19]. The methods followed to develop the ToC, are described below. The whole process was a mix of theoretical underpinnings (i.e., ToC and Systems Science), the lived experiences of the stakeholder group and the public and the expertise of the researcher, steering group and those involved in the additional focus groups.

### *2.2. Methods*

This section describes the methods followed for reviewing the systems map, the stakeholder engagement, the intervention design and the resulting theory of change.

#### 2.2.1. Reviewing the Systems Map

This step facilitated an in-depth exploration of the relationships between variables represented on the systems map. Two system science methods were chosen; causal loop construction to further explore causal relationships that had been identified within the system, and stock and flow analysis, a more detailed analysis of what has accumulated in and what flows through the system, requiring information on units or magnitudes, forcing a more rigorous analysis. The stock and flow analysis ensures that the intervention design considers contextual and pragmatic elements such as national policies and action plans and resulting funding streams, existing context and local resources. As a result, stakeholders move naturally from ideation and co-creation to a pragmatic and deliverable implementation plan. Both steps were completed by the embedded researcher and presented throughout the engagement process to inform discussions.

#### Causal Loop Construction

A causal loop diagram represents the casual relationships in a system with the direction of the relationship between variables represented by arrows. They can be reinforcing or balancing. In reinforcing feedback loops, the effect of the first variable alters the second, which feeds back to affect the first variable again, in the same direction i.e., a vicious circle. The presence of reinforcing causal loops leads to exponential acceleration over time. In balancing feedback loops, variables affect each other in opposite directions. The identification of reinforcing causal loops can suggest leverage points in a system, where

a small change in one thing can produce big changes in everything. Two reinforcing feedback loops of particular interest were identified during the community-based research and stakeholder consultation relating to school travel and car-dependency. These were explored by the researcher, facilitators and relevant stakeholders using the information gathered during these steps and with reference to the detailed causal loop diagram on car dependency in the Irish transport system [15]. Two of the most widely discussed cycles of behaviour during public engagement were school travel and car-dependency. Two causal loop diagrams were constructed by the researcher to represent these cycles of behaviour.

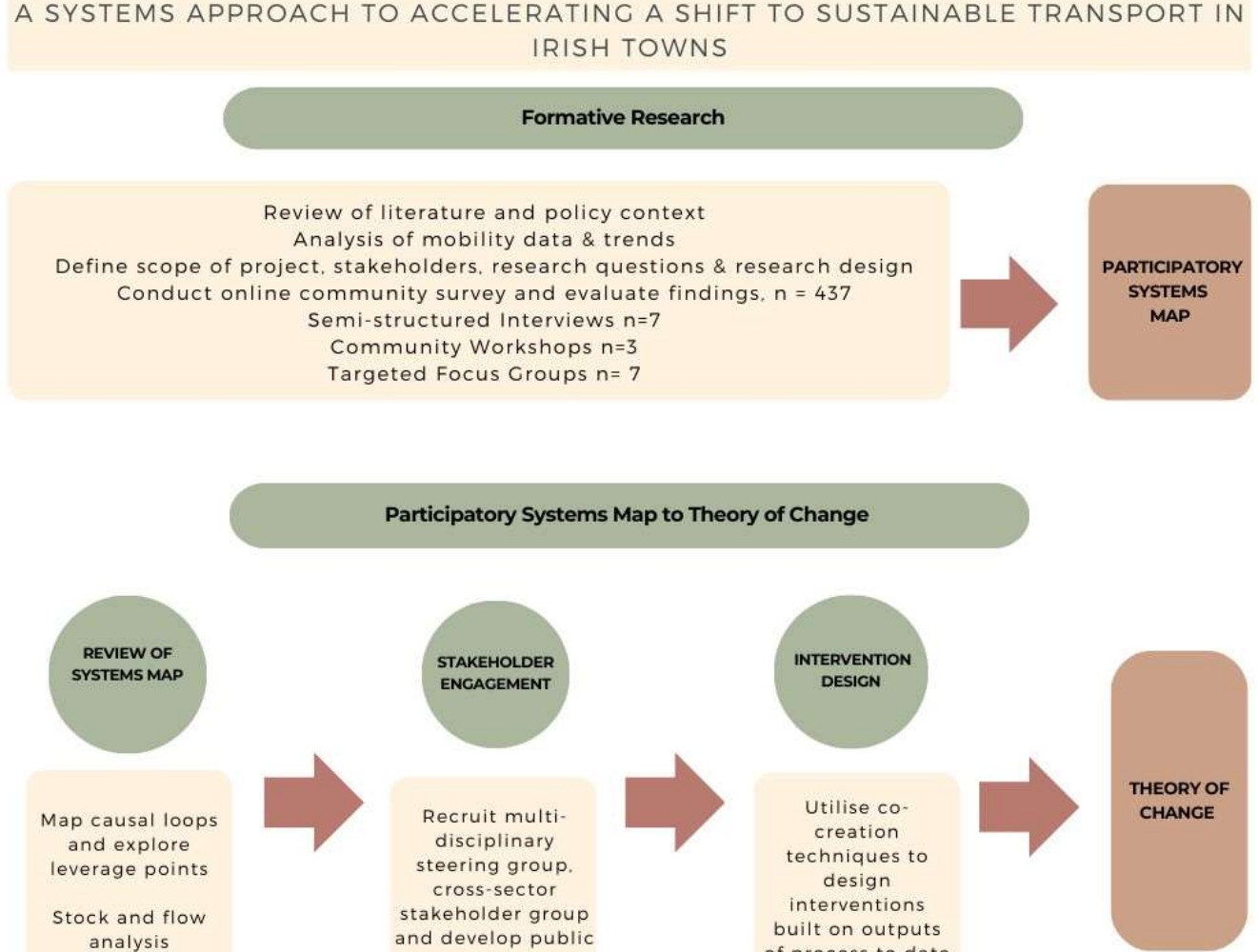

**Figure 1.** The process used for developing a systems map and theory of change framework.

Key variables in these two systems and their feedback patterns were identified from the stakeholder consultation. An online systems mapping software, Kumu.io 101 was used to display these feedback loops. The variables influencing school travel and car-dependency were represented as elements. The relationships between the variables are displayed as links. Red arrows were used to denote that one variable adds to the next, and blue arrows denote takes from. R identifies a reinforcing loop and a possible leverage point in the system.

Stock and Flow Analysis

Stock and flow analysis enables an understanding of what is already locked into the system and how this changes over time with different flows. Stock and flow analysis can help policymakers understand the potential impacts of different mobility interventions

and policy decisions. By adjusting the flow rates and promoting sustainable mobility, cities can work towards creating more sustainable mobility options. The accumulated stocks identified for consideration included land use, investment to date in transport, space allocated to modes, existing transport infrastructure and services in place and car/bicycle ownership and access. Stocks are impacted by flows, flows are what goes into and what comes out of systems. The corresponding flows are changes in land use, funding streams, reallocation of space, changes in the availability of transport modes and changes in vehicle ownership/access. Flows at the macro level may be policy-led, resulting in strategic shifts but taking place over longer time periods.

Stocks can also include more qualitative or intangible measures, such as social norms, values, beliefs or cycling competencies and are usually captured through qualitative evaluations. Although not as easily summarised over time, these are equally as important as quantitative stocks and should receive the same consideration during intervention design. The initial community engagement resulted in a rich database of information on qualitative stocks and factors impacting flows such as attitudes to mobility, barriers to cycling, perceptions of their community, place-making, working structures in the local authority and collaborations. The data were compiled from the sources specified below in Table 1.

**Table 1.** Stock and Flow Analysis.

| | Stock | Flow | Data Source |
|---|---|---|---|
| Integration of Transport and Land Use Planning | Land Use and Population Densities | Zoning Map for Development Changes in Population Densities | Kilkenny County Council Census |
| Investment to Date in Car, Public Transport or Active Travel Infrastructure | Investment to Date € million | Future Investment Patterns and Funding Streams | Kilkenny County Council Department of Transport |
| Space Allocated to Modes and Existing Transport Infrastructure | Current Allocation of Space | Planned Reallocations and New Developments | Kilkenny County Council—not currently available |
| Existing Transport Services and Usage Patterns | Rail Services Bus Services Road Infrastructure | Passenger numbers and quality of service | National Transport Authority Private Services |
| Modal Shares | Current patterns | Trends over time | Census |
| Vehicle Ownership/Access | Car/bicycle ownership Shared schemes availability Taxi Services | Changes in ownership Planned provision of schemes | Census, Commercial partners, operational reports, Department of Transport |

### 2.2.2. Stakeholder Engagement

A three-tier model was adopted for stakeholder engagement. The entire process was overseen by a steering group within Kilkenny County Council, with an emphasis on cross-sectoral and multi-disciplinary engagement to allow for communication between departments and a broad approach. This group consisted of staff members from the Local Authority; those who would have strategic oversight of the plan (Senior Management) and senior staff across departments, responsible for operationalising the plan. The steering group met prior to each stakeholder group and public consultation and reviewed the information presented at each consultation and the reports arising from the consultation.

Stakeholders were suggested by the steering group and the list was reviewed against the quintuple helix model [30] representing government, academia, industry, civil society and the environment and against the Sustainable Urban Mobility Planning Topic Guide [31]. This group met in advance of each public consultation and played a crucial role in generating a vision and designing interventions. Stakeholders were recruited by the researcher and local authority through email invitations, phone calls and in-person meetings. The wider community was involved through a hybrid approach with in-person and online events

with targeted outreach to under-represented population cohorts. Input and feedback were captured at each stage of the process.

2.2.3. Intervention Design

To facilitate the co-creation of interventions, various platforms were required to enable the public to engage with all the stakeholders, to maximise the dialogue between all, allow for informed decision-making, nurture shared learnings and arrive at local ownership of solutions generated. A facilitation team, with expertise in civil engagement, was employed to co-design the intervention design process with the researcher and the steering group. This is comprised of a series of sequential steps and is outlined below (see Figure 2). Reports were compiled by the facilitation team following each step. The full reports were made available on the project website. The reports were analysed quantitatively and qualitatively by the researcher and the facilitation team.

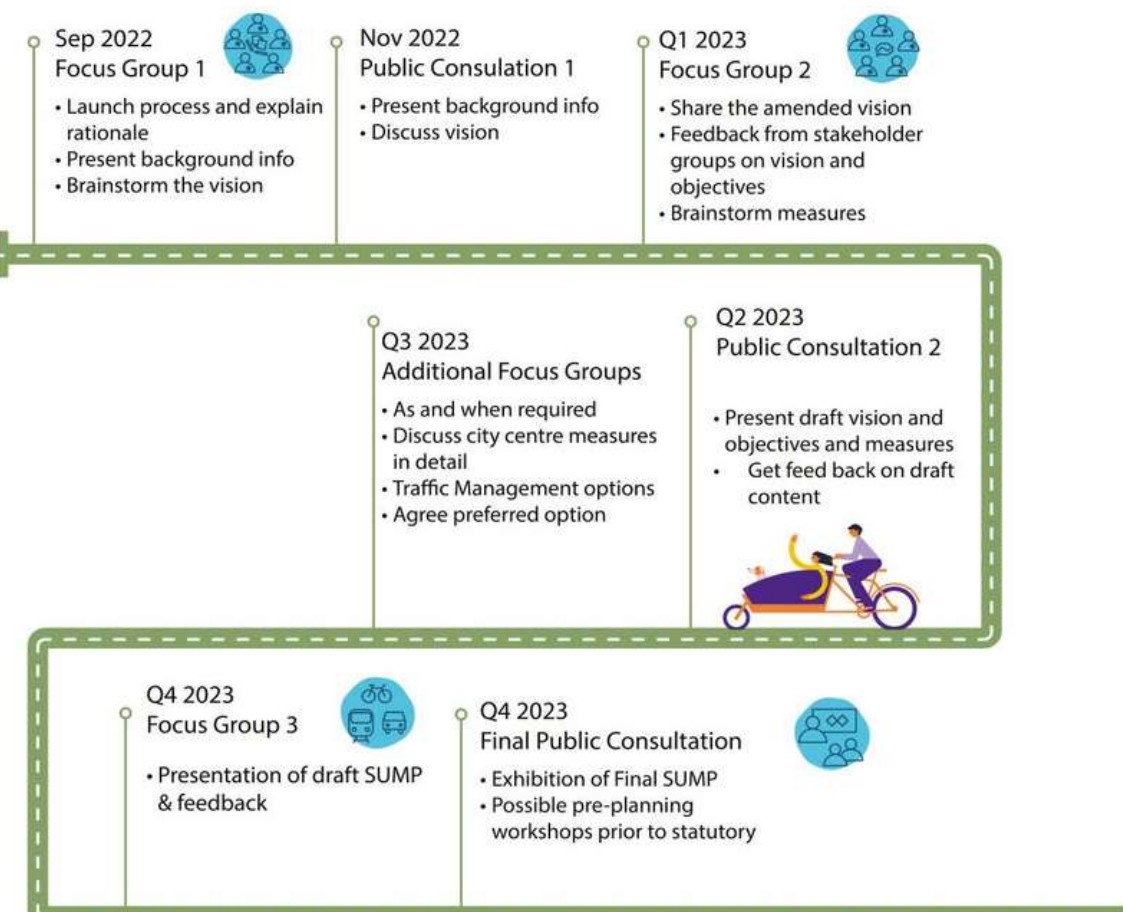

**Figure 2.** Engagement process for the development of a sustainable urban mobility plan by Connect The Dots. https://consult.kilkenny.ie/en/consultation/kilkenny-city-sustainable-urban-mobility-plan (accessed on 1 October 2023).

Stakeholder Focus Group and Public Consultation 1

The first focus group took place on the 30 September 2022. Information was presented to the focus groups from practitioners, researchers and experts in the field to enable informed decision-making as outlined in Supplementary Materials S2. Group strategies such as flexible grouping, thinking, pairing, sharing, brainstorming and group discussion were used throughout the day. The first worksheet allowed participants to present initial thoughts on presentations (surprises, interests and concerns) followed by workshops

focused on ideation. Participants were encouraged to consider their ideal city, and the challenges in moving towards the "Ideal City" and identified solutions to these challenges. Finally, they were asked to propose a future vision for Kilkenny.

The first Public Consultation was held on the 22nd of November, as an in-person event and widely advertised through social media and local media. Information from the stakeholder focus group was presented in visual format on display boards and video presentations. The presentations and reports from the stakeholder focus group were made available online. Separate sessions were held for students and the Public Participation Network, with a summary of the presentations provided by the researcher and city engineer. Similar to the stakeholder group, participants were asked to imagine their ideal city and the role that urban mobility would play in this. They considered who this 'ideal city' would be for and how it would be accessible to all. Notes were made on post-its, added to the worksheets and discussed. Over 200 people participated in the public engagement through the three forums, and feedback was gathered by a team of facilitators.

Stakeholder Focus Group and Public Consultation 2

The 2nd focus group was held on the 25 January 2023. Highlights of two recent publications were presented by the researcher, the Climate Action Plan 2023 [32] and Re-designing Ireland's Transport for Net Zero [15]. Additionally, graphs of commuter patterns, population demographics and modal shares over time were shared. The draft vision and objectives from the previous public consultation were presented. The focus group employed co-creation techniques such as scenario building and world café to explore intervention under each objective, allowing time for deliberative discussion. This information was collated by the facilitation team and presented to the steering group. The steering group reviewed the draft objectives and measures, highlighting challenges, additional opportunities and identifying underlying assumptions. A report on the draft vision and objectives was then prepared to inform the second public consultation (Report 3).

The second public engagement adopted a much broader approach to extend the reach of the engagement and was conducted over a two-month period, from April to May, in 2023. A survey was designed that presented the draft vision, objectives and suggested interventions and invited feedback on each objective and allowed respondents to rank the importance of the objectives. This was hosted online on the project website and widely disseminated through social media, advertisements on bus stops, a leaflet drop to city residents, a public display in the shopping centre, local radio interviews and three in-person events. There were 420 respondents, 39 of which were from in-person conversations at the public events. Demographic profiles were gathered from the respondents and concerns and suggestions were gathered for each objective. This information was collated by the facilitation team and a final report was drafted (Report 4).

Additional Focus Groups to Develop Theory of Change

Through the utilisation of a theory of change framework, a clear, logical flow describes how the planned interventions intend to contribute to the desired change. The mechanism of change is identified and causation is explored. Underlying assumptions, fundamental to successful outcomes are explored and risks identified, thereby ensuring a sound logic for achieving change. Targeted focus groups were facilitated by the researcher with stakeholders with relevant expertise to consider the mechanism of change for each intervention against the outputs and intended outcomes, the key risks associated with the interventions and the underlying assumptions essential for success. Furthermore, the resources required for the implementation of the intervention were reviewed and identified. Focus groups took place from May to July 2023. All focus groups were annotated and suggested interventions were categorized and incorporated into the theory of change. In this way, the findings of the systems approach were translated into a theory of change and ultimately an operational plan with actions assigned to stakeholders with the skillset and resources for implementation. To date, over 20 focus groups have been held with some

having evolved into teams to oversee the implementation of the objective and continue to meet on an ongoing basis. Different approaches were adopted when objectives required additional expertise. The reallocation of space in the city centre required traffic modelling and transport consultants were engaged for this purpose. The city centre cycle network was developed in conjunction with the Active Travel section of the National Transport Authority. The objective of rural mobility was further developed through participation in a European project on rural mobility (ongoing).

Data Analysis

The reports from the first stakeholder meeting (n = 42) and the public consultation (n = 174) were collated. An inductive thematic analysis was conducted of the dataset using the six-phase coding framework [33]. Following familiarisation with the dataset, an initial round of coding was conducted. The codes were collapsed into themes by the facilitators and researcher. The themes were presented to the steering group for discussion and a draft vision with core principles was created. The researcher then formulated the themes into a suite of objectives for review by the steering group. Following the review, the vision, core principles and objectives for the Sustainable Urban Mobility Plan for Kilkenny City were drafted for public consultation.

The second round of consultation presented the draft vision, principles and objectives to the stakeholder group initially (n = 40). Reports were compiled from worksheets, post-its and note-taking by the facilitation team, resulting in a rich dataset. Stakeholders were invited to discuss and suggest interventions under each objective. The suggested interventions were then added to the draft vision and objectives. Additional notes were made of other concerns and ideas that arose. Following this, the researcher and steering group reviewed the vision, objectives and interventions and prepared a second draft for public consultation. This round of public consultation presented the vision, objectives and interventions in a survey format, resulting in a quantitative and qualitative dataset (n = 420). The quantitative results are presented below. The qualitative data that resulted were suggestions and concerns under each objective. These were presented to the stakeholders in the additional focus groups in the final consideration of interventions. The suggestions were considered together with the guidance provided by the relevant policies, the stock and flow analysis and the causal loop construction and were incorporated into the interventions. The concerns were thematically analysed and presented as risks under the relevant objectives using the steps described above.

### 2.2.4. Theory of Change

This step entailed the detailed development of each intervention, underpinned by the ToC, by stakeholders with relevant expertise, and facilitated by the researcher. The suggestions and concerns from the engagement process formed the basis of the discussion. The mechanism of change, the outputs and intended outcomes, the key risks and the underlying assumptions were discussed and incorporated into a logical flow. During the focus groups, indicators were identified for each intervention that were measurable in the short term and of relevance to the stakeholder leading the intervention. These were cross-checked by the researcher with best practice guides. Indicators were chosen to drive stakeholder work packages and allow for regular reviews [31,34]. Future consultations will be held to present the draft Sustainable Urban Mobility Plan to the stakeholders and public for final comment. The final step will be the presentation of the Sustainable Urban Mobility Plan to the elected members of Kilkenny County Council for adoption to satisfy the statutory process.

### 3. Results

This section presents the results of the review of the systems map, the stakeholder engagement, the intervention design and the presentation of the theory of change.

### 3.1. Review of Systems Map

#### 3.1.1. Causal Loop 1: School Travel

Causal loop 1 allows us to look at school travel (Figure 3). During the engagement process, this was highlighted as a vicious circle or reinforcing feedback loop which has resulted in the greatest decline in the numbers cycling and a corresponding increase in the numbers traveling by car. The increase in car ownership in the late 1990s resulted in greater traffic volumes and a reactive increase in the allocation of space to cars outside schools. In turn, this has resulted in an unsafe environment for walking and cycling, pushing more people towards the safety of car travel and creating unsustainable growth in car travel (R1). This has been triangulated by modal shares in Kilkenny. The consequences of the unsustainable growth in car travel are an increase in carbon emissions, noise and air pollution, reductions in health and well-being, inequitable use of public space and implications for mobility justice.

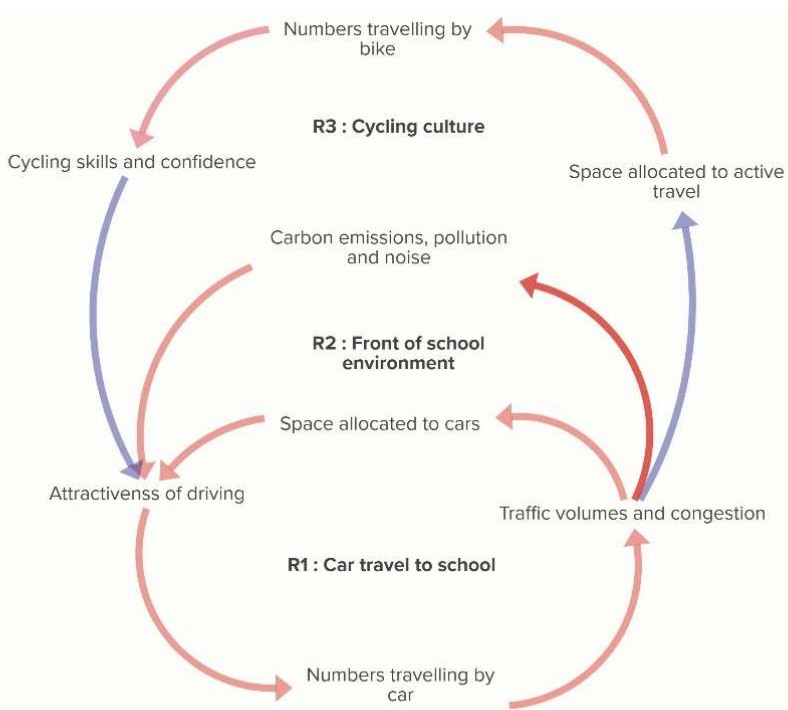

**Figure 3.** Causal loop diagram of school travel; red arrows denote adds to, blue arrows denote takes from, R identifies a reinforcing loop.

The leverage point identified below is to replace the reactive policy of space allocation to cars with a transformative policy of space allocation to active modes and constrictive measures for cars. Reallocation of space to active modes at schools and forcing cars away from the school gates will create the safe environment needed for active travel to school. Furthermore, this transformative policy will also reduce the impact of the reinforcing feedback loops, R2 and R3. The former results in increased air and noise pollution and carbon emissions and the latter results in a loss of cycling skills due to the lack of a safe environment for gaining essential skills.

#### 3.1.2. Causal Loop 2: Car Dependency

Causal loop 2 examines car dependency (see Figure 4). "Too far to travel", "no other options", and "not feeling safe" were some of the quotes continuously identifying the issue of car dependency in the community consultation. Traditional, dispersed housing patterns and a dearth of alternative modes of travel have created a dependency on cars in Kilkenny. Subsequently, there has been a prioritisation of investment in roads for cars. Ongoing investment in car infrastructure increases the attractiveness of driving. This increases traffic

volumes, leading to increased demand for road space and drives further investment in car infrastructure creating a vicious circle (R1). The attractiveness, comfort and convenience of driving, brought about by investment, results in more dispersed housing patterns and even greater transport demand (R2). Ongoing public investment in roads results in less investment in sustainable transport modes. This is compounded by the additional cost of sustainable transport provision in areas of low-density housing (R3) and further exacerbates car-dependency. The identified leverage point is to prioritise investment into sustainable modes over road capacity for cars, moving towards transit-led development.

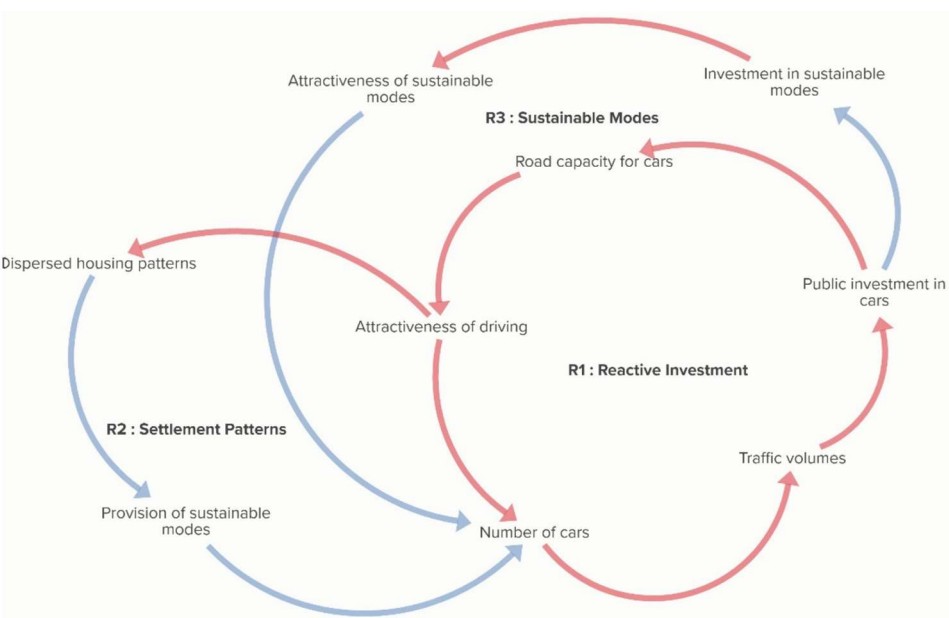

**Figure 4.** Causal loop diagram of car dependency; red arrows denote adds to, blue arrows denote takes from, R identifies a reinforcing loop.

### 3.1.3. Stock and Flow Analysis

Supplementary Materials S3 presents the full information and sources for the stock and flow analysis with a summary provided below.

Investment to date in car, public transport or active travel infrastructure: Until recently, transport investment has prioritised car design with a resulting dearth of investment in public transport and active travel infrastructure. In 2021, the Irish government introduced a 2:1 investment allocation ratio between public transport and road infrastructure: for every euro spent on road infrastructure, two are spent on public transport infrastructure. The transport budget for the period 2021–2030 allocates 46% of funding to public transport (15.7 billion euros), 11% to active modes (3.6 billion euros) and 43% to roads for car use (14.8 billion euros) [16]. Public transport is a centralised function in Ireland and is therefore not within the remit of the local authorities. However, the local authorities have dedicated active travel teams in place since 2021, with a remit of delivering active travel projects. This represents a new flow of resources, with over €3 million available in Kilkenny for active travel projects in 2022 and 2023.

Public Realm: Public realm enhancements that prioritise pedestrians, cyclists, sustainable transport and permeable spaces represent an inward flow into a system shifting towards sustainable modes. For smaller local authorities, European structural funds are an important source of funding for projects. This can lead to the regeneration or development of public spaces with an emphasis on designing for sustainable mobility and regeneration of urban centres. In the most recent applications, there is a shift from active travel to the regeneration of town centres. This will encourage infill development, but it may pose a risk to the funding available to deliver more costly, active travel infrastructure projects.

Population, housing density and zoning patterns: Low-density housing and dispersed housing patterns built up over time reinforce car-dependency and generate a long-term flow of car-centric behavioural patterns that are difficult to shift. Kilkenny City has adopted a four-neighbourhood model to accommodate expansion around the city, to continue to develop a compact urban form and to deliver the concept of the 10-Minute City and therefore reduce the overall transport demand. The two new neighbourhoods will be the focus of greenfield development and are located outside of, but immediately contiguous to the built-up area. They will cater to 70% of the housing demand with 30% catered for within the existing built-up footprint of the city, facilitated by the active land management policy, housing 4144, or 1/3 of the targeted population allocation for County Kilkenny.

Mobility infrastructure and services: A significant development in Kilkenny that may greatly alter flows in the mobility system is the introduction of a new city bus service in 2019, with two routes, every 30 min, carrying over 20,000 passengers in a 4-week period. Similarly, a bike share scheme was introduced in 2022 by Kilkenny County Council, in collaboration with a commercial partner. There are over 40 parking spaces in operation and 60 bikes deployed.

Modal share and car ownership: Without the availability of a household travel survey, modal share data from the census is the most representative data of flows over time. These data only account for the primary journey to work, school or education. Journeys to work are primarily by car, accounting for 66%, walking has fallen to 18%, with cycling just above 3%, illustrating a high dependency on car travel.

### 3.2. Stakeholder Engagement

The following stakeholders (Figure 5) were recruited as a result of the stakeholder engagement. The Stakeholder Focus group had between 40 and 50 members participating.

**STEERING GROUP**

Local Authority Management team, Representatives from Planning, Transport, Environment, Climate Action and Community

**STAKEHOLDER FOUCS GROUP**

Members of the Steering Group, Representatives from the Public Participation Network (Representatives from the Community and Voluntary section), Guards, Healthy Kilkenny, Local Development Groups, Disability Groups, Cycling Advocates, Large Employers, Traders, Older Adult Forum, Youth Representatives, Resident Associations, Arts Groups and other Community Organisations

**PUBLIC CONSULTATIONS**

**Figure 5.** Stakeholder engagement.

### 3.3. Intervention Design

Following the first round of co-creation, the dataset, incorporating two reports compiled by the facilitation team, was thematically analysed by the researcher. These reports are available on the project website https://consult.kilkenny.ie/en/consultation/kilkenny-city-sustainable-urban-mobility-plan (accessed on 1 October 2023) [35]. The dataset was coded initially and subsequently, emerging themes were identified, and are presented

below in Table 2. Together with the input of the steering group, these were reviewed and formulated into 16 objectives. The 16 objectives were presented in worksheet format at the next stakeholder focus group. This round of co-creation resulted in a comprehensive list of interventions, categorised under each objective. This was captured by the facilitators through worksheets, post-its and note-taking. This list of interventions was reviewed by the expert steering group. It was then presented for the second round of public consultation.

**Table 2.** Emergent themes, codes and their corresponding objectives.

| Theme | Codes | Objective Number |
|---|---|---|
| Integrate transport & land use planning | Encourage city centre living | 1 |
| | Rengenerate city centre buildings | |
| | Integrate sustainable transport in new neighbourhoods | |
| | Dispersed housing patterns | |
| Collaboration | Co-creation of solutions | 2 |
| | Partnerships to progress opportunities | |
| | Build trust and understanding of roles | |
| | Resistance to change | |
| | Funding opportunities | |
| | Nurture innovation | |
| Reallocate space | Less conflict with cars | 3 |
| | More space for sustainable modes | |
| | Better use of public realm | |
| | Child friendly spaces | |
| | Independent journeys for vulnerable road users | |
| | Accessibility—places for sitting and rest | |
| | Access for retail | |
| Natural Environment | Embrace the river | 3 |
| | Enhance biodiversity and greening | |
| Cycle Network | Connected cycling facilities, address severance points | 4 |
| | Safety at junctions | |
| | Perceptions of safety | |
| | Segregated facilities | |
| | Permeability and shortcuts | |
| Integration of all modes | Real time information | 5 |
| | Multi-modal hubs | |
| | Dynamic parking management | |
| | Shared bikes and car sharing options | 7 |
| | Availability of trial schemes | |
| | Integrated platform for information and booking | |
| Rural Connections | Rural public transport and connectivity | 6 |
| | Park and strides/rides | |
| | Innovative rural solutions such as community car pools | |
| Public Transport in the City | Directness and frequency | 6 |

**Table 2.** *Cont.*

| Theme | Codes | Objective Number |
|---|---|---|
| | Additional routes | |
| | Improved connectivity | |
| Greenways | Greenways and attractive off-road spaces for cycling | 8 and 9 |
| | Cycle tourism | |
| Community Wide Programmes | Learn to cycle courses | 10 |
| | Communication campaigns | |
| | Social norms and habits | |
| | Workplace campaigns | |
| | Cycle hubs for visible presence | 11 |
| Safe Routes to School | Safe infrastructure in school vicinities | 12 |
| | Campaigns in schools | |
| | Education in schools | |
| Vibrant City | Health, air quality, noise pollution | Vision |
| | Economically thriving | |
| | Local producers and services | |
| | Engage with tradition of arts and culture | |
| | Create social spaces | |
| | Preserve heritage | |
| | Place-making, attractive, inclusive spaces | |
| | Connectivity within the city and the region | |

The survey format utilised in the public consultation elicited qualitative and quantitative data. The quantitative data presented a snapshot of the views of the community on the process to date (n = 420). Respondents ranged in age from between 16 to 65 and above. 26% were aged between 35–44 and a further 25% were aged between 45–54. 50% of the respondents lived in Kilkenny City and a further 33% lived in the county. Over 50% chose the city centre as their area of work. When asked about how they travel to or from Kilkenny City, 50% travelled by car or van, 6% by bus and 2% by train. 24% walked or used a wheelchair and 14% cycled. For journeys within the city, 41% walked, 38% travelled by car and 5% travelled by bus. Respondents rated the vision and each objective on a 5-star rating scale, with 5 being the highest. 60% of respondents rated the vision 5 out of 5. The reallocation of space in the city centre was ranked as priority one by 55%, with pedestrianisation or partial pedestrianisation being the favoured option. The ratings of all other objectives ranged from 3.98 to 4.32 with Safe Routes to School receiving the highest rating. The enhancement of public transport was also highly ranked at 4.16 out of 5. This objective elicited over 141 comments, over half of which related to extended routes and stops and greater frequency of service.

### 3.4. Theory of Change

The additional focus groups centred on the co-creation of the theory of change with the relevant stakeholders and researchers. The mechanism of change, outputs and outcomes for each intervention were worked through in a logical sequence and mapped under its objective. Funding sources and stakeholders with responsibility for delivery were noted. Where possible the stakeholders with responsibility for delivery were present at the workshop. Indicators of success and risks to successful implementation were identified. The resulting theory of change

is presented in Supplementary Materials S4 with a concise table presented below (Table 3). The indicators relating to the objectives are presented in Figure 6.

**Table 3.** Theory of Change to Reduce Car-Dependency and Create a Shift to Sustainable Transport in Kilkenny City.

| Interventions | Mechanism of Change | Long Term Outcomes |
|---|---|---|
| **CREATE SUSTAINABLE SYSTEMS**<br>**Objective 1: Integrate transport and land use planning** | | |
| Review **policies** and data **systems** in place<br>Regulate for **compact growth**, the **10 min city** and city centre living. | **Reshaping environment** through prioritisation of sustainable transport infrastructure<br>Informed decision making | Reduction in travel demand and reduced car dependency<br>Appropriate investment in cycling infrastructure |
| **Objective 2: Encourage multi-sectoral engagement** | | |
| **Partnership** work with other cities<br>Establish **multi-sectoral** teams and encourage **citizen engagement**<br>Establish pre-planning communication channels/forums | Greater **awareness** within the community of proposed developments<br>Shared learnings, **motivations** and actions | Effective, sustainable partnerships in place<br>Cohesive decision making and strong civic and social communities<br>Designs that reflect the needs of the community |
| **DESIGN HEALTHY BUILT ENVIRONMENTS**<br>**Objective 3: Reallocate and prioritise space in urban centres for pedestrians, cyclists, sustainable transport options, public uses and green spaces** | | |
| **Reallocation** of space to prioritise pedestrians, cyclists, collective transport, public uses & green spaces | **Redesign** of urban spaces | Enhanced liveability of urban centres & use of public realm<br>Sustainable urban centres |
| **Objective 4: Design and build a strategic cycling network** | | |
| **Connect** population & workplace densities, daytime populations, retail, health & amenity services | **Facilitation** of modal shift through **provision of infrastructure** | Reduced car dependency, reduced emissions, greater levels of physical activity and well being |
| **Objective 5: Integrate all modes and develop wayfinding, legibility and MAAS** | | |
| Develop a parking management strategy and **multi-modal hubs** to **integrate** all transport modes<br>Use smart, real time applications, enhance **wayfinding** | **Use restrictive measures** to **discourage** use of the car<br>**Facilitate** end user through enhanced provision, access to and clarity of **information** | Reduction in car dependency<br>Greater mobility choices |
| **Objective 6: Strengthen and improve city, inter city and rural public transport links** | | |
| Adequate public transport and **Park and Stride/Ride/Pedal** provision<br>**Rural connectivity** with community and **demand responsive transport** | **Facilitation** of modal shift through **provision of public transport options** | Reduction in car dependency & sprawl<br>Rural nodes with greater access to services<br>Reduced emissions, greater levels of physical activity and well being |
| **Objective 7: Develop micro-mobility and car sharing options** | | |
| Provision of micro mobility, **car sharing, e-bikes, assisted trikes, mobility scooters** & other mobility solutions | **Facilitation** of modal shift through **provision** of micro-mobility and car sharing options | Reduced car dependency, reduced emissions, greater mobility choices |
| **Objective 8: Develop greenway/off road facilities to encourage inexperienced/learner cyclists** | | |
| **Identify opportunities** for development of greenway facilities, conduct feasibility, design & deliver | **Provision** of safe environment for inexperienced cyclists<br>**Encouraging** growth in cycling tourism and industry | Greater number of citizens and tourists cycling |



**Table 3.** *Cont.*

| Interventions | Mechanism of Change | Long Term Outcomes |
|---|---|---|
| **ENGAGE SOCIETY** | | |
| **Objective 9: Promote cycling as a tourism offering** | | |
| Promote **suitable routes** with on road and off road offerings | Greater awareness of opportunities Provision of safe environment Modelling and persuasion | Greater number of tourists attracted by cycling offering |
| **Objective 10: Promote cycling through community wide programmes** | | |
| Develop a delivery mechanism and provide an **offering of cycling education courses** | Education and training, development of skills, building confidence and competency | Greater number of cyclists in the community |
| **Objective 11: Develop cycling hubs** | | |
| Identify and develop **cycling hubs** in the community | **Facilitate growth** of cycling community Facilitate roll out of **innovative initiatives** | Social conditions that support cycling in the community |
| **Objective 12: Develop safe routes to school and behaviour change campaigns for school travel** | | |
| Develop **safe, active travel routes** to school **& park & stride** arrangements Discourage car traffic at the **front of school environment** Promote active travel to school through **education and awareness** | **Facilitation** of modal shift through provision of safe routes to school Education and training, development of skills, building **confidence** and **competency** | Greater modal share of walking and cycling to school Improved cycling confidence and skills in the community |
| **EMPOWER PEOPLE** | | |
| **Objective 13: Provide focused training and education programmes for those with lower levels of participation in cycling** | | |
| Develop **delivery model** for cycling education courses Provide **targeted training** for those with additional barriers | Education and training, development of skills, building confidence and competency | Greater number of cyclists in the community |
| **Objective 14: Enhance attractiveness and comfort of sustainable routes using place-making techniques** | | |
| **Engage** with local communities to enhance **attractiveness** of routes & public spaces through placemaking | **Engagement** and **capacity building** | Enhanced liveability and social capital |
| **Objective 15: Develop and deliver communication campaigns to promote a shift to sustainable transport** | | |
| Develop **campaigns** to increase awareness of **benefits**, provide information on **opportunities** & **encourage** a shift to sustainable transport modes | **Awareness raising**, **information provision** and **encouragement** | Greater number of people using sustainable transport options in the community |
| **PRIORITISE ROAD SAFETY** | | |
| **Objective 16: Enhance safety of all road users through road design and reduction in traffic volumes, speeds, % of HGVs** | | |
| Reduce traffic volumes through **traffic management plans** Introduce **traffic calming**, **additional crossings**, **pedestrian** & **cyclist treatment** and **junction tightening** at all junctions | Provision of safe infrastructure to **facilitate** active modes Enhanced **perception** of **safety** | Enhanced **safety** of all road users Greater number of less confident cyclists using cycle routes **Reduction** in **collisions**, **injuries** and **deaths** |

Create Sustainable Systems, Design Healthy Built Environments, Engage Society, Empower People and Prioritise Road Safety

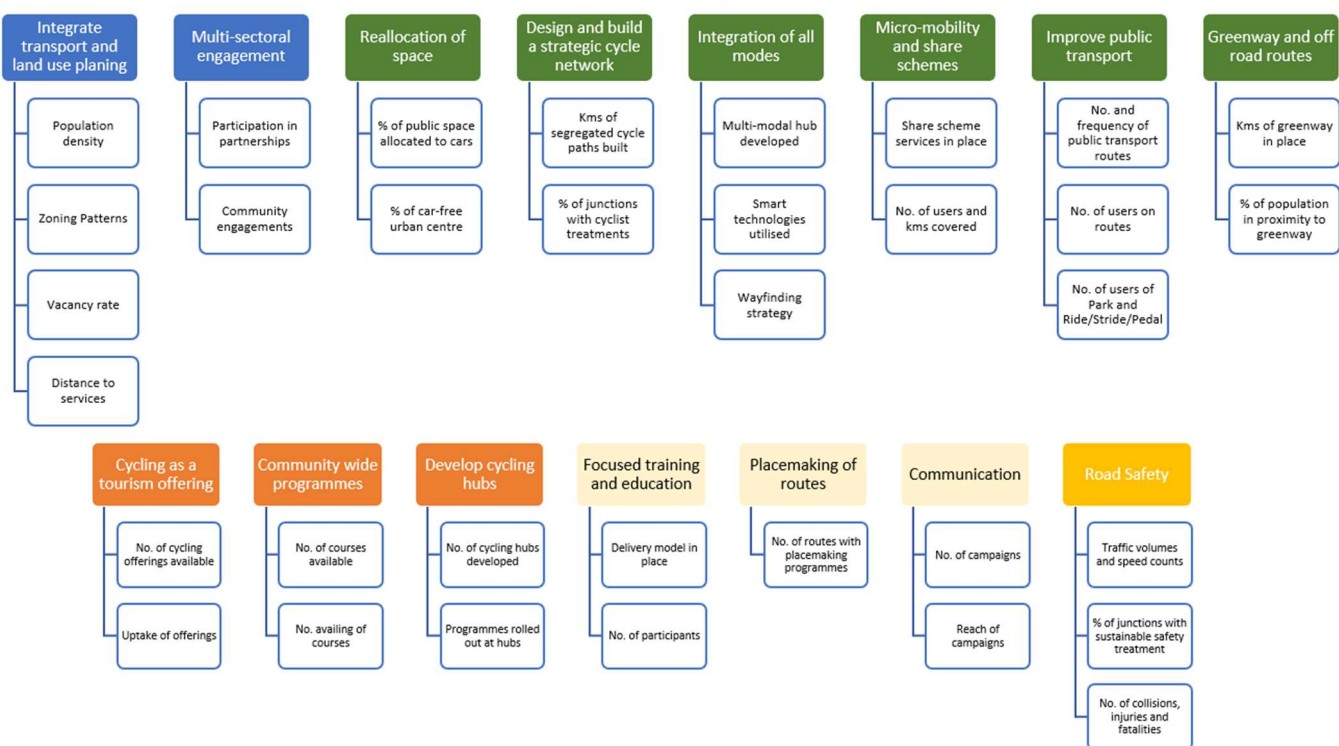

**Figure 6.** Objectives and Indicators for the Theory of Change.

## 4. Discussion

This research, conducted over a three-year period, describes the process based on systems science that was adopted to develop a theory of change to accelerate a shift to sustainable transport in Kilkenny City, Ireland. Specifically, this paper details the series of steps taken to translate the process of participatory systems mapping into a pragmatic framework. Causal loop construction and stock and flow analysis were utilised to inform decision-making during the co-creative process of intervention design. This led to a theory of change with sixteen key objectives with a logical flow from inputs to desired outcomes. This template can be utilised by other small cities or towns to enable a shift to sustainable transport modes.

Critical to the success of this process was the establishment of a platform of decision-makers and stakeholders with the remit to act on all leverage points in the system. A joined-up approach allows for greater awareness of transport, land use and health challenges in urban centres. Collaborative work nurtures idea generation, greater understanding of the roles of others, and decisions informed by the needs of the communities. In line with other studies conducted in small cities, the participation and co-responsibility of public and private stakeholders are the cornerstone of successful Sustainable Urban Mobility Plans [36].

The lengthy engagement process undertaken as part of this research prioritised informed decision-making and presented alternative solutions to the stakeholders and opportunities for debate. In doing so, a much greater understanding of entrenched car dependency in Ireland [14] was gained and a wider breadth of interventions were generated. This is reflected in the first key objective in the theory of change; integrate transport and land-use planning and selected indicators, such as densities, zoning patterns and distances to services. The importance of strengthening town centres through densification and regeneration was widely acknowledged. This is mirrored in the literature; similarly to

larger cities, avoiding development in the outer part of cities and towns and regenerating centres is advised to reduce car dependency and encourage sustainable modes [37,38].

Throughout the engagement process, it was widely recognised that the target of reducing transport emissions by 50% by 2030 on 2018 levels [32] would only be met with systemic changes. The stakeholders largely concurred that to harness the investment that has already accumulated in the transport system and to ensure the attractiveness and liveability of the town centre, the guiding principle to achieve modal shift should be the reallocation of existing road space, Objective 3, to sustainable modes and car-free urban centres. This is consistent with the conclusions of the most recent policy review [15]. However, the reallocation of space still generates fear of a negative impact on the accessibility of the city centre to visitors, rural dwellers and nearby residents and subsequent disruption to and/or closures of businesses. In Kilkenny, almost 70% of the population are rural dwellers with high levels of forced car dependency, and a lack of public transport options. For those living within 2 km of the city centre, there is an absence of safe cycling facilities and a limited city bus service. Similarly, for those with reduced mobility, there is a gap in alternative transport modes that may result in a loss of access to the city in the event of reallocation of space away from cars. These fears and concerns were echoed across public consultations, stakeholder meetings and in the political discourse. Similar to other case studies [39], these concerns are foremost with traders but shared by the stakeholders and community. Recent studies have shown that business sentiment is shaped by lived experiences and improvements in process or design-related elements can generate buy-in [40]. This was echoed by stakeholders, citing the importance of the "wow factor", or urban architecture when transforming spaces. The enhancement of public transport options both within and to the city was seen as a pre-requisite to any car demand management measures. To cater to the rural population, the need for park and rides/strides and pedals and the integration of multi-modality solutions and share schemes was a strong recurring theme. With the stakeholders and the public, there were concerns that this may not be deliverable as it is not within the remit of the local authority and national leadership and investment were required. As a smaller city, Kilkenny will struggle to compete for this investment. However, examples were provided from other small European cities of shared e-cars for people with mobility impairments, mini-bus sharing and micro-mobility solutions. Ongoing engagement with European programmes will ensure that decision-makers in Kilkenny will be poised to trial innovative solutions.

The completion of a safe, segregated, connected cycle network and an enhanced pedestrian environment was considered a necessity to enable a shift to sustainable modes and is well recognised in reviews of cycle network planning [41]. At the time of the project, the government announced a 20% allocation of the transport budget for active travel, €365 million per annum. With the policies and funding now in place, doubts remained over the level of expertise nationally to deliver cycling projects, uncertainty around the availability of and adherence to design guidance, a lack of appetite and political support for tackling hard decisions and historic work structures and patterns favouring car-centric environments. There were worries about public opposition to road space reallocation and the removal of car parking for the provision of cycling facilities. In agreement with the literature [42], the stakeholders concurred that a multi-faceted approach was required to cultivate a cycling culture in Kilkenny, to ensure that the benefits of a shift to cycling would be community-wide. The importance of empowering people to shift to cycling through the provision of off-road facilities, cycle education programmes, cycle hubs, bike share schemes and targeted education is captured in the theory of change.

Notably, for smaller cities, work commutes are often longer, from more varied origins and beyond the reach of active travel [38]. Therefore, trips to non-work destinations, such as services, recreation and shopping, should be a key consideration. In Kilkenny, school travel was considered a priority as a high percentage of these existing journeys are within active travel distance. A reduction in companion journeys to school has great potential to reduce or eliminate circuitous car journeys, reduce congestion at key travel times, provide

health benefits [43,44] and reduce the burden of care and encourage independent travel at a young age [45]. Secondary school journeys (12 to 18 years) may prove more challenging to shift to active travel due to longer journey distances and additional barriers, carrying items and social norms [46]. Tourism is another consideration for smaller cities, and in line with other studies [47,48], there are opportunities to promote cycling as a tourism offering. Coupled with the presence of shared schemes, improved public transport and demand-responsive transport, cycling can be part of the solution to car-free access to tourist destinations and sustainable tourism.

This process has resulted in a theory of change to reduce car-dependency and create a shift to sustainable transport modes, in Kilkenny City. The sixteen objectives identified above will provide a framework for small cities and towns, which can be further adapted to the local context through a co-creation process as described above. The engagement of external facilitators can augment the transparency and creativity of the process and avoid consultation fatigue. The process requires a longer planning stage, in this case, up to 18 months, with rigorous preparation and presentation of material to inform decision-making and be transparent through the stages. Although resource-intensive, the use of a systems approach and systems science techniques combined with the theory of change framework results in a stakeholder engagement process that fosters collaboration, trust and inclusive decision-making. Shared learning is embedded in the progression from consultation, to co-creation of interventions, to the identification of behaviour change mechanisms and finally a roadmap for implementation. The process will result in a framework that considers the mobility system as a whole, the resources available locally and the travel nuances of the community. Furthermore, the intended outcomes will be embedded in all organisations involved. The risks identified for implementation of the plan are similar to other cities; lack of funding and human resources to deliver actions, political challenges, lengthy planning processes, entrenched social norms and resistance from the community [49].

The next stage of this process will action the interventions, under each objective in the theory of change. The targets and key performance indicators will be operationalised and monitoring systems will be embedded into normal practice. Open source tools and methods will be reviewed and applied [50], enabling the ongoing monitoring of the Sustainable Urban Mobility Plan for Kilkenny City to ensure that transport planning decisions promote health and wellbeing, address transport inequities and environmental concerns, and move Kilkenny towards a net zero, healthy and sustainable city.

## 5. Conclusions

This work synthesises the scientific evidence, a participatory systems approach, and the practical application of the embedded researcher resulting in a pragmatic framework to reduce car-dependency and create a shift to sustainable travel modes. The co-created theory of change will form the basis of a Sustainable Urban Mobility Plan for Kilkenny. It will inform the prioritisation of work packages and will result in an agile, iterative programme of work that will change over time as the selected key performance indicators are reviewed. Fundamental to the process was the early engagement of stakeholders and the community and a multi-disciplinary steering group from the local authority, resulting in a shared vision and ownership of the plan. The design of the process must provide space for deliberative debate, learning and innovation. The inclusion of input from expert panels throughout the engagement process nurtured shared learnings and informed decision-making, garnering community and political support. The theory of change presented is a template that can now be more easily adapted to new settings by the participatory processes outlined.

**Supplementary Materials:** The following supporting information can be downloaded at: https://www.mdpi.com/article/10.3390/su151914633/s1, Supplementary Materials S1–S4, including the systems map, the information presented at the focus groups and public meetings, the stock and flow analysis and the complete theory of change.

**Author Contributions:** Conceptualization, B.L. and N.M.; Methodology C.C., B.L. and N.M.; Formal analysis, C.C.; Writing—original draft preparation, C.C.; Writing—reviewing and editing, C.C., B.L. and N.M.; Supervision, B.L. and N.M. All authors have read and agreed to the published version of the manuscript.

**Funding:** This research was funded by the Irish Research Council, grant number EBPPG/2021/96.

**Institutional Review Board Statement:** This study was conducted in accordance with the Declaration of Helsinki, and approved by the Ethics Committee of Waterford Institute of Technology (WIT2021REC014), 1 June 2021.

**Informed Consent Statement:** Informed consent was obtained from all subjects involved in the study.

**Data Availability Statement:** The raw data supporting the conclusions of this article will be made available by the authors, without undue reservation. Reports on all consultations are on the website https://consult.kilkenny.ie/en/consultation/kilkenny-city-sustainable-urban-mobility-plan (accessed on 1 October 2023).

**Conflicts of Interest:** The authors declare no conflict of interest.

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
