# Peer review of "Harnessing Systems Science and Co-Creation Techniques to Develop a Theory of Change towards Sustainable Transport"

_sustainability, doi:10.3390/su151914633_

Round 1

Reviewer 1 Report

The paper presents a comprehensive methodology to address the shift towards sustainable transport modes through co-creation techniques. The topic is of importance in the context of transitioning to sustainable mobility, and the authors provide valuable insights into the challenges and opportunities associated with designing local solutions to this aim. The paper is well-written and structure. I recommend the publication of this work with once some minor modifications are made:

1.       Section headings should be clearer. Some headings are repeated, and section numbering seems to not have been formatted properly.

2.       Table 2 and Figure 6 are not formatted properly, with no right margin.

3.       Appendix figures do not have y axis labels.

Reviewer 2 Report

This paper starts with a thorough and interesting introduction. It is clearly a valid and topical research topic in promoting sustainable transport, but in its current form the paper not suitable for an academic journal article.

It is trying to cover far too much and is lacking in focus. There is not a clear aim or objective. The Abstract states that “this study expands on the systems approach” which is far too loose and clearly follows on from a previous paper (Corry et al., 2023). Straight-away, the title, abstract and introduction section are far too long. Some of the introduction should have been a literature review.

The title shows a methodological focus on systems science and co-creation techniques. Then there is the broad and vague Theory of Change, which has a reasonable introduction but it is not clear at the end of the paper how this paper has provided a theoretical contribution. I find behavioural psychology theories such as Theory of Planned Behaviour, Contemplation of Change, together with market segmentation approaches, more useful in understanding the promotion of sustainable transport and the types of people who might change. I remain convinced by the Theory of Change.

There is a valid and appropriate case study of the Irish town of Kilkenny, but the paper seems more policy focused than academic. It reads more like an EU project report than a journal paper, which I suspect it came from.

The paper is far too descriptive, laboriously going through stages 6 – 10 of an overall project. Table 2 (or is it 3?) stretches over too many pages (ten in total). When writing I would try and keep a table on one page of A4, may be two at most.

There is no analysis in the paper, qualitative or quantitative. The Table is presented as de facto findings.

I look for 2-3 key take-aways from a journal paper. There is nothing really coming out of this paper. The Conclusions section is far too brief. The Discussion section is an interesting read but not relevant.

The paper is generally well written with useful and relevant references, but the authors should note the signposting of the sub-headings is wrong in the middle of the paper (all as 1.1 or 1.1.1).

Reviewer 3 Report

I had fun reading the article, the topic is very interesting and necessary.  I have some suggestions to make regarding the article:

- In the abstract I would clarify the method used

- At the end of the introduction I would clarify the different sections in which we are going to develop the article.

- I think it is necessary to justify the method used, as well as to propose other methods that we could use in the research.

- It would be useful to refer to other cities that meet the same conditions as Kilkenny, to check if there are similar studies and to make a comparison of results.

- On the other hand, it would be useful to assess the amount of investment involved in this project and to determine a possible opportunity cost.

Congratulations and good work.

Reviewer 4 Report

In the article, the authors considered sustainable transport from the perspective of long-term ecological and economic stability, in line with the sustainability trend, and proposed a concept of changes in the mobility system. Having read the content of the study, I suggest supplementing the content in the subsections in order to improve the manuscript in accordance with the following suggestions:

1. The abstract should specify the purpose of the article.

2. In the introduction, it should be indicated why the topic taken up by the authors is important and how the conducted research broadened the knowledge on the issues of sustainable transport.

3. The article should be supplemented with a chapter related to the literature review, shortening the introductory information.

4. In methodology 1.1. The research project should accurately describe the date of conducting the research, characterize the survey questions and the research sample.

Minor typos in the text should be corrected

Round 2

Reviewer 2 Report

Much of the paper has been improved, which is pleasing to see, and the authors have responded well to my queries. It is good to see more of a focus on the theory of change.

However, I still have a major concern about the lack of analysis. Only stock and flow analysis is covered in the paper. The response to my review question was: “All focus groups, workshops and consultations were thematically analysed.  The presentation of the analysis is beyond the scope of the paper. The reports are all available on the project website and the key findings have been included in the results section”.

I would still like some of this thematic analysis included in the paper, say a few paragraphs in the Methods section and a couple of pages in the Results section. The paper still seems incomplete to me.

There are some other minor comments on the paper.

The title is still too long. Too many “and”s for me.

The Abstract also seems a bit too long for me.

The presentation of Table 2 is really poor in the version I have. I can’t properly read it.

The page numbering system does not work and starts again.

There is a lack of clarity with Level 1 and Level 2 headings, particularly where there is Methods and Methodology Level 1 headings.

Round 3

Reviewer 2 Report

The authors have responded to my revision requests and all six have been satisfactorily addressed. I am therefore able to recommend publication.